Awareness of security and privacy settings in video conferencing apps among faculty during the COVID-19 pandemic

Alammary Ali 1
http://orcid.org/0000-0002-2895-3482 Alshaikh Moneer 2
http://orcid.org/0000-0003-0372-5851 Pratama Ahmad R. 3 ahmad.rafie@uii.ac.id
1 College of Computing and Informatics, Saudi Electronic University , Riyadh , Saudi Arabia
2 Department of Cybersecurity, College of Computer Science and Engineering, The University of Jeddah , Jeddah , Saudi Arabia
3 Department of Informatics, Universitas Islam Indonesia , Sleman, DI Yogyakarta , Indonesia
Cherukuri Aswani Kumar
Electronic publication date: 2022 Jul 7
Publication date: 2022
Volume: 8
Electronic Location ID: e1021
Received 2022 Mar 16; Accepted 2022 Jun 6
Copyright: © 2022 Alammary et al.
Copyright year: 2022
Copyright holder: Alammary et al.
License: This is an open access article distributed under the terms of the Creative Commons Attribution License, which permits unrestricted use, distribution, reproduction and adaptation in any medium and for any purpose provided that it is properly attributed. For attribution, the original author(s), title, publication source (PeerJ Computer Science) and either DOI or URL of the article must be cited.
License URL: https://creativecommons.org/licenses/by/4.0/

Keywords: COVID-19 pandemic, E-learning, Online learning, Video conference, Faculty members, Security awareness, Security policy, Privacy concerns, Security and privacy settings, Digital literacy

Funding: Deputyship for Research & Innovation, Ministry of Education 8002 This work was supported by the Deputyship for Research & Innovation, Ministry of Education in Saudi Arabia through the project number 8002. The funders had no role in study design, data collection and analysis, decision to publish, or preparation of the manuscript.

==============================
COVID-19 has forced many schools and universities worldwide, including Saudi Arabia, to move from traditional face-to-face learning to online learning. Most online learning activities involve the use of video conferencing apps to facilitate synchronous learning sessions. While some faculty members were not accustomed to using video conferencing apps, they had no other choice than to jump on board regardless of their readiness, one of which involved security and privacy awareness. On the other hand, video conferencing apps users face a number of security and privacy threats and vulnerabilities, many of which rely on human factors to be exploited. In this study, we used survey data from 307 faculty members at 43 Saudi Arabian universities to determine the level of awareness among Saudi Arabian faculty regarding security and privacy settings of video conferencing apps and to investigate the factors associated with it. We analyzed the data using the Knowledge-Attitudes-Behaviors (KAB) model and the Partial Least Squares Structural Equation Modeling (PLS-SEM) method. We found that the average awareness score of video conferencing apps’ security and privacy settings falls into the “Poor” category, which is not surprising considering that many faculty members only started using this new technology on a daily basis because of the pandemic. Further analysis showed that perceived security, familiarity with the app, and digital literacy of faculty members are significantly associated with higher awareness. Privacy concerns are significantly associated with higher awareness only among STEM faculty members, while attitudes toward ICT for teaching and research are negatively associated with such awareness among senior faculty members with more than 10 years of experience. This study lays the foundation for future research and user education on the security and privacy settings of video conferencing applications.

Introduction

During the COVID-19 pandemic, video conferencing apps have gained popularity as the most viable way to facilitate business processes during work-from-home or learning-from-home strategies. According to a cross-country study (Pratama, 2017), there is an intriguing bidirectional relationship between ICT adoption and education indicators, and video conferencing apps are no exception. In many educational institutions, including those in Saudi Arabia, they have become the most popular tool for supporting online learning (Alshehri et al., 2020). Due to their unique characteristics, they are ideal for use in teaching and learning (Alammary, Carbone & Sheard, 2016). They enable instructors to set up online synchronous classes that can be recorded and accessed at a later time, either for those who want to review the class again or for those who missed a class. Using personal computers, laptops, or even mobile phones, students can attend these online classes from anywhere (Camilleri & Camilleri, 2021).

Even prior to the COVID-19 pandemic, a variety of video conferencing applications were available. Many other video conferencing apps, such as Google Meet, Microsoft Teams, and Blackboard Collaborate Ultra, also saw an increase in downloads and usage as a result of the COVID-19 pandemic (Trueman, 2020). The vast majority of universities in Saudi Arabia use Blackboard Collaborate Ultra because the Ministry of Education (MOE) has designated the Blackboard platform as the official e-learning platform (Rahmatullah, 2021). Prior to the COVID-19 pandemic, the majority of Saudi Arabian faculty were unfamiliar with Blackboard Collaborate Ultra; however, they had no choice but to adopt it regardless of their level of preparedness (Alammary, Alshaikh & Alhogail, 2021).

On the other hand, the shift from traditional face-to-face to online learning that occurred during the pandemic has raised numerous cybersecurity and protection of individual and organizational information resource concerns (Almaiah, Al-Khasawneh & Althunibat, 2020). Several cybersecurity and privacy threats and vulnerabilities plague users of video conferencing apps, including the exposure of user data, unwanted and disruptive intrusions, the spread of malware, and the hijacking of host machines through remote control tools (Joshi & Singh, 2017). Recent security reports indicate that the number of cyberattacks against numerous organizations, including universities, has increased significantly (Hakak et al., 2020; Lallie et al., 2021). Many cybersecurity incidents are caused by employees who disregard security policies due to their initial lack of security awareness (Hina & Dominic, 2016). Due to their role as instructors, who must frequently administer and host online classes on their own, faculty members become the primary actors in higher education. As a result, assessing their knowledge of security and privacy settings when using video conferencing apps is crucial for ensuring that everyone’s online learning experience is as secure and private as possible.

This is the first study to use a national survey to assess the security and privacy awareness of Saudi Arabian faculty members regarding the use of Blackboard Collaborate Ultra as the primary video conferencing app during the COVID-19 pandemic. The specific objectives of this study are to (1) comprehend and investigate the factors associated with the Saudi Arabian faculty’s level of security and privacy awareness regarding the use of video conferencing apps, particularly Blackboard Collaborate Ultra, which is the most widely used video conferencing app in this country for teaching and research purposes during and possibly beyond the COVID-19 pandemic, and (2) assist universities, particularly in Saudi Arabia, in improving their security policies.

Online learning in Saudi Arabian universities during the COVID-19 pandemic

In 2005, the Saudi National E-learning Center (NELC) was established. In Saudi Arabia, NELC is in charge of establishing governance frameworks and regulations for e-learning and online learning. NELC plays a significant role in enhancing the online learning experience in universities and promoting effective online learning practices (Alqahtani & Rajkhan, 2020). In addition, the center develops policies and procedures for the delivery and administration of online learning programs. The policies stipulate the technologies that universities must implement and the level of support they must provide for their faculty and students. Policies also include standards and practices for creating online learning environments that are accessible (National eLearning Center, 2021).

NELC has provided universities with guidelines that specify e-learning infrastructure including hardware (e.g., servers, storage, and networking), e-learning solutions (e.g., learning management systems and video conferencing apps), establishing dedicated deanships to manage e-learning matters, providing training and awareness programs, and other e-learning and online learning initiatives (Malik, Faiza Abid & Bhatti, 2018). As a result of NELC’s efforts and investment in e-learning, universities in Saudi Arabia have a sufficient online learning infrastructure (Thorpe & Alsuwayed, 2019). During the COVID-19 pandemic, other researchers discovered that the IT infrastructure in Saudi Arabian universities successfully supported the transition from face-to-face to online learning (Alqahtani & Rajkhan, 2020). However, the maturity level of the university played a significant role in its ability to overcome obstacles and implement e-learning solutions before the pandemic.

Research indicates that faculty and students in Saudi Arabia have overwhelmingly positive attitudes towards e-learning (Hoq, 2020). Others have discovered that Saudi Arabian students prefer e-learning due to its adaptability and enhanced communication with their teachers and peers. However, the same study revealed that students viewed online instruction as less advantageous than traditional face-to-face instruction (El-Sayed Ebaid, 2020). While students’ attitudes toward e-learning are partially influenced by their prior experience and readiness for online learning (Alqahtani, Innab & Bahari, 2021), numerous other factors, including gender, course level, and quality of online learning approaches, also play a role (Elumalai et al., 2020).

In the midst of the COVID-19 pandemic, which has presented significant challenges to societies, Saudi Arabia, along with numerous other nations, has attempted to adapt to the impending crisis. During the COVID-19 pandemic, Saudi Arabian universities took a number of steps to accommodate the decision to adopt online learning, as education was one of the sectors most severely impacted by the pandemic. To support and facilitate the transition to e-learning and online learning, the Ministry of Education (MOE) of Saudi Arabia has provided all universities with e-learning solutions licenses, including Blackboard Learning Management System (LMS) and its video conferencing app, Blackboard Collaborate Ultra (Ministry of Education, 2021). In addition to providing free internet access to students across the nation, the Ministry of Education increased bandwidth to accommodate the high demand for internet connections. In collaboration with charitable organizations, the Ministry of Education has also provided laptops and training to deserving students (Alqahtani & Rajkhan, 2020).

As of May 2020, the MOE reports that approximately 1.6 million students have taken more than four million online exams at 43 private and public universities. Over 58,179 faculty members participated in this transition by delivering lectures, administering exams, and holding online discussions, averaging 1.5 million online classes per week (Ministry of Education, 2021). UNESCO has lauded Saudi Arabia’s efforts to pursue successful transitions to online learning to facilitate the education of over 6 million students in schools and universities during the COVID-19 pandemic (Elumalai et al., 2020). For these transitions to be successful, it is essential that faculty members serving as frontline instructors are able to provide students with the best online learning experience possible. This includes ensuring that their awareness level is sufficient to keep online learning activities secure and private for all parties.

Theoretical framework

In 2006, Kruger & Kearney (2006) developed the KAB model, a prototype for assessing information security awareness, which consists of three different dimensions (i.e., knowledge, attitudes, and behaviors). Each dimension is measured by a series of multiple-choice questions with either correct or incorrect answers for all three dimensions, as well as a “Don’t know” option for the “knowledge” and “attitudes” dimensions only. Since then, this KAB model has been widely adopted as a tool for evaluating information security awareness (McCormac et al., 2017; Onumo, Ullah-Awan & Cullen, 2021).

In addition, based on our literature review, we identified a number of individual factors that are associated with security and privacy awareness. We identified five specific factors to include in this study: attitudes toward information and communication technology (ICT) for teaching and research, digital literacy, privacy concerns, perceived security awareness, and familiarity with the video conferencing app platform.

Perceived security policy awareness and privacy concerns

There is a large body of literature on people’s concerns about how their private information is shared when they use information technology products and services (Petronio & Child, 2020), as well as on how they perceive themselves to have the security awareness to protect them (Bulgurcu, Cavusoglu & Benbasat, 2010; Li et al., 2019). According to these studies, perceived security policy awareness and privacy concerns result in more cautious decisions regarding the use of information technology-related goods and services. Consequently, we anticipate the same for the video conferencing applications investigated in this study. The first two hypotheses in this study are therefore: H1: Faculty with a higher level of perceived security policy awareness have a higher level of awareness of video conferencing apps’ security and privacy settings.

H2: Faculty with a higher level of privacy concerns have a higher level of awareness of video conferencing apps’ security and privacy settings.

Attitudes toward ICT for teaching & research and digital literacy

Indisputable is the contribution of ICT to the improvement of education. However, it is no secret that not all faculty share the same attitudes toward the use of ICT for teaching and research, either because of their lack of experience, which may translate into lower levels of digital literacy (Cavas et al., 2009), or because of their personal preferences (Bauwens et al., 2020). Taking these findings into account, we hypothesize that: H3: Faculty with more positive attitudes toward ICT for teaching and research have a higher level of awareness of video conferencing apps’ security and privacy settings.

H4: Faculty with a higher level of digital literacy have a higher level of awareness of video conferencing apps’ security and privacy settings.

Familiarity with the app

In the wake of the global adoption of distance learning in response to the COVID-19 pandemic, numerous institutions use a variety of video conferencing apps. Each app may have its own user interface and minor features, despite the fact that the majority of video conferencing apps will have the same major features. Consequently, we hypothesize that: H5: Faculty who are more familiar with the video conferencing app in use have a higher level of awareness of video conferencing apps’ security and privacy settings.

Based on all the aforementioned hypotheses, the conceptual model awareness in video conferencing apps’ security and privacy settings in this study is depicted in Fig. 1.

Figure 1 Conceptual model of video conferencing security and privacy settings awareness in this study.

Materials and Methods

Target population

There are 29 public universities and 14 private universities in Saudi Arabia (Ministry of Education, 2020). Due to the spread of the COVID-19 pandemic, the Saudi government has suspended face-to-face instruction in all of these universities as of the middle of the spring 2020 semester. The Ministry of Education demanded that universities move all courses online utilizing the available e-learning solutions. Online education will continue for the following three semesters. This study’s target population consisted of professors from any Saudi Arabian university who were teaching during the study period. Included in this category are teaching assistants, lecturers, assistant professors, associate professors, and full professors. According to the Ministry of Education’s most recent report, published in 2020, there were approximately 71,000 faculty teaching in Saudi Arabian universities (Ministry of Education, 2021). Universities in Saudi Arabia typically provide their faculty with computers, assign them email addresses, and require them to regularly check their emails. Consequently, it can be stated that the entire population of interest to this study was theoretically accessible.

Measures

There are five latent exogenous variables in this study (i.e., attitudes toward ICT for teaching and research, digital literacy, perceived privacy concerns, perceived security awareness, and familiarity with Blackboard Collaborate Ultra). As summarized in Table 1, we created, adopted, or adapted items from other studies for all five. In addition, three other observed variables (i.e., knowledge, attitudes, and behaviors) will be merged into a single composite score to answer the first research question and will be treated as a latent endogenous variable (i.e., awareness of security and privacy settings on Blackboard Collaborate Ultra) to answer the second research question.

Table 1 Research instruments for all latent independent variables.

Variables	Measurement items	Code	Source	
Perceived Security Policy Awareness	I know the rules and regulations prescribed by the security policy of my university.	pspa1	(Bulgurcu, Cavusoglu & Benbasat, 2010)	
I understand the rules and regulations prescribed by the security policy of my university.	pspa2	
I know my responsibilities as prescribed in the security policy to enhance the information security of my university.	pspa3	
Privacy Concerns	User online privacy is really a matter of users’ right to exercise control and autonomy over decisions about how their information is collected, used, and shared.	pc1	(Malhotra, Kim & Agarwal, 2004)	
I believe that online privacy is invaded when control is lost or unwillingly reduced as a result of a marketing transaction.	pc2	
Companies seeking information online should disclose the way the data are collected, processed, and used.	pc3	
It is very important to me that I am aware and knowledgeable about how my personal information will be used.	pc4	
When online companies ask me for personal information, I sometimes think twice before providing it.	pc5	
I’m concerned that online companies are collecting too much personal information about me.	pc6	
Attitudes toward ICT for Teaching & Research	I like using ICT for teaching or conducting research.	att1	(Ng, 2012)	
ICT makes teaching or conducting research more interesting.	att2	
I am more motivated to teach or to conduct research with ICT.	att3	
There is a lot of potential in the use of mobile technologies (e.g., smartphones, tablets) for teaching and research.	att4	
Digital Literacy	I know how to solve my own technical problems.	dl1	(Ng, 2012)	
I can learn new technologies easily.	dl2	
I know about a lot of different technologies.	dl3	
I am confident with my search and evaluate skills in regard to obtaining information from the Web.	dl4	
I am familiar with issues related to web-based activities e.g., cyber safety, search issues, plagiarism.	dl5	
ICT enables me to collaborate better with their peers on project work and other learning activities.	dl6	
I frequently obtain help with my university work from my friends over the Internet e.g., through email, social media, or Videoconference.	dl7	
Familiarity	How familiar are you with Blackboard Collaborate Ultra?	fam1	Developed by the authors	
How much do you use Blackboard Collaborate Ultra?	fam2	
Have you read the terms & agreements of Blackboard Collaborate Ultra?	fam3	

Perceived security policy awareness

We adopted the scales from Bulgurcu, Cavusoglu & Benbasat (2010) to measure perceived security policy awareness among the faculty in this study. We omitted the first three out of six items in the original scales because they measured security awareness in general, as opposed to awareness of security policy within an organization, which was the focus of the last three items that we kept. The scales were measured on a five-point Likert scale.

Privacy concerns

We adopted the scales from Malhotra, Kim & Agarwal (2004) to measure privacy concerns among the faculty in this study. Specifically, we selected only the two most pertinent items from three to four original items for each of the control, awareness of practice, and collection dimensions to make it six, as opposed to 10 items in total. The scales were measured on a five-point Likert scale.

Attitudes toward ICT for teaching and research

We adapted the scales from Ng (2012) to measure the attitudes of the faculty in this study toward using ICT for teaching and research. Specifically, we merged the teaching and research components into a single item, reducing the total number of items from eight to four. The scales were measured on a five-point Likert scale.

Digital literacy

We adopted the scales from Ng (2012) to measure digital literacy of the faculty in this study. In particular, we eliminated three of the original six items for the technical dimensions while retaining both items for the cognitive and social-emotional dimensions, reducing the total number of items from nine to six. The scales were measured on a five-point Likert scale.

Familiarity with blackboard collaborate ultra

We developed three items to measure the faculty’ familiarity with Blackboard Collaborate Ultra. On a five-point Likert scale, the first two questions assessed respondents’ familiarity with and frequency of use of Blackboard Collaborate Ultra, while the last questioned whether or not they had read the terms of service.

Awareness of security and privacy settings on blackboard collaborate ultra

We developed five questions about Blackboard Collaborate Ultra to assess our participants’ knowledge, attitudes, and behaviors regarding Blackboard Collaborate Ultra’s security and privacy settings. The first three questions contain side-by-side images of the app’s default and altered security and privacy settings. We asked respondents to indicate which one was the default (i.e., knowledge), which one they preferred (i.e., attitudes), and which one they primarily utilized (i.e., behaviors). For the final two questions, we presented them with two hypothetical scenarios involving security and privacy incidents and asked them to describe the available options and the actions they would take in each scenario. For each response, we assigned a score of 0 for any incorrect answer for the knowledge dimension or the worst option for the attitudes and behavior dimensions, a score of 5 for “Don’t know” or a partially correct response, and a score of 10 for the correct answer or the most secure option. The complete questions are available in Appendix 1.

Data collection

In preparation for the data collection phase, a research ethics application was submitted to the University of Jeddah ethics committee, which granted this study an ethics approval number (UJ-REC-021). We collected the data by distributing a series of questionnaires using the online survey software Qualtrics. On the survey’s landing page, we provided an explanation of the study’s purpose. In addition to being informed of the length of time required to complete the survey, participants were asked for their permission to participate. They were informed that participation is voluntary and that they may opt out or withdraw at any time. To eliminate the possibility of bias, no gifts or incentives of any kind were promised to the participants.

The questionnaire contained three sections. The first objective was to collect participant demographic information, including gender, age, level of education, academic fields, academic rank, teaching experience, and university name. This component served as a control variable for the analysis.

The second section consisted of Likert-type questions with five levels for participants to indicate their attitudes toward ICT, their digital literacy, their perceived privacy concerns, their perceived privacy awareness, and their familiarity with Blackboard Collaborate Ultra. This section was used to measure the model’s exogenous variables.

In the third section, we measured the actual security and privacy awareness score of each participant, which represented their awareness of security and privacy settings on Blackboard Collaborate Ultra and served as the endogenous variable in our model. Four scenarios were included. Each scenario included: (a) two screenshots captured from Blackboard Collaborate Ultra either during a running session or from the settings window; and (b) several questions designed to assess the knowledge, attitudes, and behaviors of each participant regarding some important security and privacy settings in Blackboard Collaborate Ultra, as well as potential security and privacy issues that may arise while using the application.

The objective of the first scenario was to assess participants’ awareness of the risks associated with guest access. The second scenario involved the activation of vital permissions, such as media file sharing and whiteboard access. The third scenario involved private chat rooms that could be abused to disseminate harmful and inappropriate content. The final scenario was intended to assess participants’ knowledge of what to do when malicious links are posted in chat.

Before we began collecting data, we conducted a pilot study to confirm the content validity of the survey items, evaluate their difficulty, and obtain rough estimates of the time required to complete the survey. Validity was evaluated based on its content and appearance. Validity of content can help “establish an instrument’s credibility, accuracy, relevance, and breadth of domain knowledge.” Face validity, on the other hand, examines survey items for “ease of use, clarity, and readability” (Burton & Mazerolle, 2011). Twelve faculty members from multiple universities in Saudi Arabia were invited to participate in the pilot study. The pilot survey was created as an online survey using Qualtrics survey software. Participants were given a blank text field to comment on the item’s relevance and clarity. In addition, they were required to indicate whether a revision was necessary for the item. Participants were encouraged to provide a revision for the item if one was suggested. All participant feedback was analyzed, and a few survey questions were modified to reflect the suggested alterations. Several items were rewritten to improve their relevance or clarity. Others were eliminated due to irrelevance or duplication. In addition, items were added to measure certain missing aspects.

The survey was then distributed to faculty members at Saudi Arabia’s 43 public and private universities. To reach a representative sample, invitations were emailed, and the survey link was shared on the four most popular social media platforms in Saudi Arabia: WhatsApp, LinkedIn, Telegram, and Twitter. On these platforms, the objective was to reach as many academic communities as possible. The responses were gathered between October and November of 2021. Around 470 responses were received, and 307 were determined to be complete and valid. The vast majority of responses that were eliminated were incomplete, but there were also a small number of participants who refused to participate in the survey.

Data analysis

To determine the actual level of security and privacy awareness on video conferencing apps among Saudi Arabian faculty in this study, we calculated a composite score with a 3:2:5 weight as suggested by Kruger and Kearney for each of knowledge, attitudes, and behaviors variable based on the responses to the third survey part (Kruger & Kearney, 2006). The final security and privacy awareness score for each participant was then normalized to a range between 0 and 100. According to Kruger and Kearney’s classification, a score below 60 is categorized as “Poor,” a score of 80 or higher is categorized as “Good,” and a score in between is categorized as “Average” (Kruger & Kearney, 2006).

We employed the Partial Least Squares Structural Equation Modeling (PLS-SEM) technique with the “plssem” package in STATA 15 to identify factors associated with the awareness of video conferencing apps’ security and privacy settings (Venturini & Mehmetoglu, 2019). PLS-SEM is a widely employed structural equation modeling approach that permits the estimation of complex relationships between latent variables in path models. It is especially useful for the exploration and development of theory, as well as when prediction is the primary purpose of a study, and it performs well with a small sample size (Hair, Howard & Nitzl, 2020). Prior to running the path analysis, we examined the standardized loadings of each measurement item to identify any that should be excluded from the model, and we used the squared interfactor correlation and average variance extracted (AVE) from the path analysis to confirm discriminant validity. To address the issue of endogeneity, we employed the control variable strategy by conducting multiple multigroup comparison analyses across all demographic variables (Hult et al., 2018). The complete code and data set can be found in our GitHub repository.

Results

Participant demographic information

The demographic information of all participants in this study is summarized in Table 2. In terms of gender (50.81% male vs. 49.19% female), academic fields (46.91% STEM vs. 53.09% non-STEM), and position, the sample is quite balanced (57.33% tenured vs. 42.67% non-tenured). The majority of participants are under 45 years old (80.7%), hold a PhD (60.5%), and have at least 10 years of academic experience (63.84%).

Table 2 Demographic information of participants.

Variable	Categories	Freq	%	
Gender	Male	156	50.81	
Female	151	49.19	
Age	<35	87	28.34	
35–44	161	52.44	
45–54	52	16.94	
55–64	7	2.28	
Education	Bachelor’s degree	9	2.93	
Master’s degree	112	36.48	
PhD degree	186	60.59	
Rank	Teaching assistant (Nontenured)	23	7.49	
Lecturer (Nontenured)	108	35.18	
Assistant Professor (Tenured)	135	43.97	
Associate Professor (Tenured)	33	10.75	
Full Professor (Tenured)	8	2.61	
Experience	<3 years	50	16.29	
3–9 years	146	47.56	
10–24 years	97	31.60	
25+	14	4.56	
Academic Fields	Arts, Humanities and Social Sciences	118	38.44	
Science, Engineering and Technology	144	46.91	
Medical and Health Sciences	45	14.66	

Awareness of video conferencing apps’ security and privacy settings

Figure 2 depicts the distribution of the composite score for awareness of the security and privacy settings of video conferencing applications. According to Kruger and Kearney’s classification, the overall score for all participants in this study (M = 44.27, SD = 16.06) falls into the “Poor” level of awareness (Kruger & Kearney, 2006).

Figure 2 Density plot of the video conferencing apps security and privacy settings awareness score among all participants.

Factors associated with awareness of security and privacy settings

Several measurement items in our model, all from the exogenous variables, were found to have a low standardized loading score, and thus had to be eliminated (Hair et al., 2012). After multiple iterations of this validity test for the measurement model, 17 items with standard loading scores exceeding 0.800 are retained. Table 3 provides the summary statistics for all measurement items and the results of their validity tests. Importantly, as summarized in Table 4, the AVE scores for each latent variable are greater than the squared interfactor correlation scores, confirming that the model’s discriminant validity has been met (Hair, Howard & Nitzl, 2020; Venturini & Mehmetoglu, 2019).

Table 3 Summary statistics and validity tests of the measurement model.

Variables	Measurement items	Mean	SD	Standardized loading (Initial)	Standardized loading (Final)	Cronbach	DG	rho_A	
Perceived Security Policy Awareness	pspa1	3.48	1.07	0.949	0.949	0.947	0.966	0.948	
pspa2	3.47	1.07	0.970	0.970	
pspa3	3.61	1.06	0.934	0.934	
Privacy Concerns	pc1*	4.21	0.86	0.680	–	–	–	–	
pc2	4.30	0.80	0.728	0.846	0.772	0.866	0.790	
pc3	4.53	0.73	0.714	0.802	
pc4	4.66	0.63	0.767	0.831	
pc5*	4.45	0.81	0.690	–	–	–	–	
pc6*	4.28	0.87	0.663	–	–	–	–	
Attitudes toward Using ICT for Teaching & Research	att1	4.42	0.82	0.878	0.892	0.884	0.928	0.891	
att2	4.26	0.87	0.884	0.902	
att3	4.19	0.90	0.886	0.907	
att4*	4.18	0.93	0.699	–	–	–	–	
Digital Literacy	dl1*	3.85	0.92	0.671	–	–	–	–	
dl2	4.21	0.77	0.783	0.823	0.787	0.875	0.796	
dl3	3.90	0.87	0.809	0.863	
dl4*	4.22	0.82	0.757	–	–	–	–	
dl5	3.98	0.86	0.771	0.823				
dl6*	4.26	0.79	0.765	–	–	–	–	
dl7*	3.99	0.96	0.438	–	–	–	–	
Familiarity with Blackboard Collaborate Ultra	fam1	3.87	1.18	0.857	0.914	0.741	0.884	0.764	
fam2	4.25	1.19	0.790	0.865	
fam3*	2.32	1.51	0.542	–	–	–	–	
Awareness of Security and Privacy Settings	k	3.74	0.80	0.937	0.937	0.913	0.945	0.918	
a	3.82	0.77	0.942	0.942	
b	2.79	0.69	0.889	0.889	
Note:

An asterisk (*) indicates items omitted from further analysis.

Table 4 Discriminant validity.

Variable	PSPA	PC	Att	DL	Fam	AoSPS	
PSPA	0.905						
PC	0.026	0.683					
Att	0.065	0.077	0.810				
DL	0.218	0.077	0.219	0.700			
Fam	0.034	0.008	0.012	0.069	0.792		
AoSPS	0.750	0.058	0.072	0.326	0.280	0.852	
Note:

Diagonal elements in bold are average variance extracted (AVE), off-diagonal elements are squared interfactor correlation.

The results from PLS-SEM demonstrated a very high average R2 value of 0.91, indicating a substantial coefficient of determination (Hair, Ringle & Sarstedt, 2011), as well as a relative goodness-of-fit (GoF) value of 0.99, which meets the rule of thumb (Henseler & Sarstedt, 2013; Vinzi, Trinchera & Amato, 2010). In addition, the path analysis demonstrated that all five hypothesized relationships are statistically significant. One unexpected result is the negative correlation between attitudes toward ICT for teaching and learning and awareness of the security and privacy settings of video conferencing applications. Our subsequent multigroup comparison analysis revealed that, with the exception of two demographic factors, the results are quite consistent across the board. First, the structural effect of privacy concerns on the awareness of the security and privacy settings of video conferencing apps is significant only for STEM faculty and not for non-STEM faculty. Second, the negative structural effect of attitudes toward ICT for teaching and learning on awareness of the security and privacy settings of video conferencing apps is significant only for those with more than 10 years of academic experience and not for those with less.

Based on the findings, Fig. 3 illustrates the final model of the awareness of video conferencing apps’ security and privacy settings among Saudi Arabian faculty in this study, while Table 5 summarizes the model’s hypothesis tests results.

Figure 3 Final model of video conferencing security and privacy settings awareness in this study.

Table 5 Hypothesis test results.

Hypothesis	Relationship	Result	
H1	PSPA → Awareness	Significantly positive effect	
H2	PC → Awareness	Significantly positive effect, moderated by academic fields	
H3	Att → Awareness	Significantly negative effect, moderated by teaching experience	
H4	DL → Awareness	Significantly positive effect	
H5	Fam → Awareness	Significantly positive effect	

Discussion

This study investigates Saudi Arabian faculty’s awareness of security and privacy settings of video conferencing apps, especially Blackboard Ultra that they use to teach during the COVID-19 pandemic. One of the study’s key findings is that, in general, faculty in Saudi Arabia still have poor security and privacy awareness on video conferencing apps. This may be understandable considering that most of them only got into this new technology on a daily basis because of the pandemic (Alammary, Alshaikh & Alhogail, 2021). As evidenced by earlier studies, the use of Blackboard and subsequentially Blackboard Collaborate Ultra in Saudi Arabia was relatively low prior to the pandmic (Al Meajel & Sharadgah, 2018; Tawalbeh, 2017). This finding is consistent with a previous study’s conclusion that general awareness of cybersecurity practices in Saudi Arabia is still below the desired level (Alammary, Alshaikh & Alhogail, 2021). Therefore, additional efforts must be made to raise awareness by implementing multiple strategies, not only to raise awareness but also to establish a cybersecurity culture (Alshaikh, 2020). Establishing support groups called “cyber champions” to raise academic privacy awareness and influence faculty behavior toward adopting cybersecurity practices is an example of a proposed strategy to build a cybersecurity culture. Multiple studies have found that using support groups and peers to modify cybersecurity behavior is an effective method (Alshaikh & Adamson, 2021; Cram, D’Arcy & Proudfoot, 2019; Guo et al., 2011). Universities could employ this strategy to educate their academic community about the significance of maximizing the security and privacy settings within video conferencing apps.

The effect of perceived security policy awareness on awareness of security and privacy settings in video conferencing apps is the strongest among the five latent exogenous variables examined in this study. This particular discovery is not unexpected. Despite the fact that this is not always the case (Tariq, Brynielsson & Artman, 2014), a higher perceived level of security awareness typically results in enhanced security and privacy practices (Lebek et al., 2014; Pratama & Firmansyah, 2021; Hwang et al., 2021). Therefore, it is only natural that faculty with a greater understanding of their organization’s security policies will be more willing to invest time in learning the security and privacy settings for any app they use, including video conferencing apps.

Familiarity with the video conferencing app itself was discovered to have the second largest positive effect on awareness of the security and privacy settings of the app. Given the abundance of available applications, each with its own features and configurations, familiarity with the app in use is crucial, as predicted when developing a new construct for this variable. Consequently, those unfamiliar with the video conferencing app in use may be unaware of all available security and privacy settings. This finding is also consistent with findings from other studies regarding familiarity with concepts, technical terms, or security-related systems that can aid in increasing individuals’ security awareness (Schmidt et al., 2008; Zwilling et al., 2022). Fortunately, this is a relatively straightforward issue to address, for example with adequate technical support to educate faculty on the subject.

Digital literacy was found to have a moderately positive impact on the awareness of security and privacy settings in video conferencing applications. Individuals with a higher level of digital literacy are able to independently navigate all security and privacy settings, according to a straightforward explanation. As a result, individuals with a higher level of digital literacy tend to have a heightened security and privacy awareness, which is consistent with the findings of a number of previous studies (Sasvári, Nemeslaki & Rauch, 2015; Nemeslaki & Sasvari, 2015; Cranfield et al., 2020).

Privacy concerns were identified as the fourth exogenous variable positively influencing the awareness of security and privacy settings in video conferencing applications. This result is consistent with the extensive literature on the relationship between privacy concerns and security awareness in general (Chung et al., 2021; Siponen, 2001). Unlike the previous three exogenous variables, however, this variable has a significant positive effect only on STEM faculty. In other words, the effect is essentially nonexistent among faculty members in the social sciences, arts, humanities, health, and medical sciences. This result could be explained by the increased familiarity of STEM faculty members with ICT in general, including its benefits and risks. According to a separate study, STEM faculty were more aware of cybersecurity threats like phishing than non-STEM faculty (Diaz, Sherman & Joshi, 2020). Consequently, they are more aware of the security and privacy settings of all applications, including video conferencing apps, than their non-STEM counterparts.

Unlike the other factors, attitudes toward ICT for teaching and research were found to have a negative impact on video conferencing app users’ awareness of security and privacy settings. The fact that this finding is significant only among participants with more than 10 years of teaching experience may provide insight into an intriguing story. The senior faculty appear to be accustomed to whatever ICT solutions they utilized prior to the COVID-19 pandemic, which were probably not video conferencing applications. Having to learn something new in order to perform a task they are already very familiar with may be preventing them from fully utilizing this new technology, particularly in terms of its security and privacy settings. After all, resistance to change, whether among individuals in general (Audia & Brion, 2007), or among faculty in particular (McCrickerd, 2012; Tallvid, 2016) is not something new. The good news is that despite being statistically significant, this variable has the smallest effect of all exogenous variables in this model. As a result, focusing on the other variables that have a positive effect may be enough to compensate for it.

Other than academic field and teaching experience, no statistically significant differences in age, gender, educational attainment, or tenure-track status were observed. On the one hand, this suggests that the endogeneity issue is largely addressed in this model, which strengthens the reliability of the findings. On the other hand, the relatively similar scores across demographic groups can be viewed as a positive development. It indicates equality in terms of security and privacy awareness among Saudi Arabian faculty, as is also the case in some countries (Hadlington, Binder & Stanulewicz, 2020; Pratama, Firmansyah & Rahma, 2022), whereas some other countries continue to demonstrate inequality (McCormac et al., 2017; Zwilling et al., 2022).

Conclusions

In this study, we used the Knowledge-Attitude-Behavior (KAB) model to determine the actual security and privacy awareness of faculty in Saudi Arabia regarding Blackboard Collaborate Ultra, the most widely used video conferencing app in the country, in terms of video conferencing apps. We discovered that the average score falls into the “Poor” category (mean = 44.27, SD = 16.06), which is not surprising considering that many of them only use this new technology on a daily basis because of the pandemic.

In addition, based on the results of the subsequent PLS-SEM analysis, we found that all five latent variables in our model have significant relationships with Saudi Arabian faculty members’ awareness of the security and privacy settings of video conferencing apps. In particular, perceived security policy awareness has the greatest impact, followed by familiarity with the video conferencing app’s platform and digital literacy. Moreover, perceived privacy concerns are only significant among STEM faculty, and surprisingly, attitudes toward the use of ICT for teaching have a significant, albeit small, negative impact on senior faculty with more than 10 years of teaching experience.

This study lays the groundwork for future research and interventions that aim to increase user awareness of security and privacy concerns when using video conferencing apps for educational and research purposes. Given the rapid adoption of video conferencing apps as a result of distance learning in response to the COVID-19 pandemic, it is becoming increasingly important to address this issue. Blackboard Collaborate Ultra, despite being the most applicable in the Saudi Arabian context, is not one of the most widely used video conferencing applications in the world. There are alternative applications such as Zoom, Google Meet, Microsoft Teams, Skype, and VooV Meeting among others. There may be some technical differences in their security and privacy settings, so some of the findings of this study may not necessarily be applicable to the other apps, despite the fact that their primary functions are typically identical. Therefore, we recommend that similar research be conducted in other regions of the world to account for cultural and technical differences that may make users of video conferencing apps less aware of their security and privacy settings.

Supplemental Information

Supplemental Information 1 Dataset & STATA Code for PLS-SEM Analysis.

Click here for additional data file.

Supplemental Information 2 Part 3 of the survey used to measure security and privacy awareness score.

Click here for additional data file.

Supplemental Information 3 Codebook for categorical data in the dataset.

Click here for additional data file.

We would like to thank Dr. Firman M. Firmansyah for his contributions to the research design, specifically for suggesting and reviewing the adequacy of our use of psychometric scales in developing the questionnaire for this study, as well as for copy editing the final manuscript.

Additional Information and Declarations

Competing Interests

Author Contributions

Ethics

Data Availability

The authors declare that they have no competing interests.

Ali Alammary conceived and designed the experiments, performed the experiments, authored or reviewed drafts of the article, and approved the final draft.

Moneer Alshaikh conceived and designed the experiments, authored or reviewed drafts of the article, and approved the final draft.

Ahmad R. Pratama conceived and designed the experiments, analyzed the data, performed the computation work, prepared figures and/or tables, authored or reviewed drafts of the article, and approved the final draft.

The following information was supplied relating to ethical approvals (i.e., approving body and any reference numbers):

The University of Jeddah granted Ethical approval to carry out this study (UJ-REC-021).

The following information was supplied regarding data availability:

The survey data and code in STATA format are available in the Supplemental Files and at GitHub: https://github.com/ahmadrafie/bbultrasecprisettings.

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
