# Peer review of "Awareness of security and privacy settings in video conferencing apps among faculty during the COVID-19 pandemic"

_PeerJ Computer Science, doi:10.7717/peerj-cs.1021_

## Round 0.1 · original submission · Major Revisions

Please address the reviewer's comments, in particular reviewer 3. You need to justify the proposal and prove its validity of the proposal. Further, the logical flow of the presentation and articulation of the ideas should be coherent. Please note that revision of the manuscript does not guarantee the acceptance until the reviewers agree to the corrections carried out. So please do the corrections carefully and submit the list of corrections carried out against each of the suggestion.

Reviewer 1 ·

Basic reporting

This study investigated security and privacy awareness among academics in Saudi Arabia
relating to the use of video conferencing apps.

Authors theoretical claim seems to be acceptable but there is no novelty in the work. The authors try to replicate the same story throughout the paper.

Experimental design

A survey experiment research design is employed in this paper

Validity of the findings

There are few experimental findings in the paper and it is not sufficient. Need more experimental analysis on the data.

·

Basic reporting

No comment

Experimental design

Since video conference apps are common among all kinds of professions and why this survey is targeted at academicians?
What is the logic behind the calculation of the security awareness score?
Is this study proposes any particular application as a safer one out of this study?

Validity of the findings

The considered sample size should have been higher to come to one conclusion.
Possibility of having high bias due to the non respondents and sample size
age group wise findings should have been included
What was the duration of data collection? Collection through WhatsApp, LinkedIn, Telegram, and Twitter is a viable way to reach academicians?
What is being considered as Non-STEM category as Math and Science also considered in STEM category

Additional comments

Results based discussion should be much more elaborated based on the findings
The metrics used should be properly explained
Sample questionnaire can be included

·

Basic reporting

This study examines the factors that influence academics' actual awareness of security and privacy in relation to videoconferencing apps. The data are based on a recent survey and PLS-SEM is used. This is a timeless study and SEM is appropriate. Some results are not plausible. For example, the results that perceived security awareness is negatively related to actual security and privacy awareness are a bit strange. The document needs improvement in all aspects. Below there are a number of comments that must be addressed
Main comments
Introduction: Please try to improve the description of the purpose of the research. Please add one sentence on the data and methods used (PLS-SEM). Please also clearly state the contribution of the research. Is it the first study on privacy and security issues of video conferencing apps?
Conceptual background: Section 2 “e-learning in Saudi Arabia” should be improved. The paragraph on internet speed does not fit. Section: Video conferencing in online learning. Please cite more recent references. Video conferencing apps greatly improved. References from 1976 are not helpful here.
The theoretical background must be improved: Privacy issues cannot be explained by privacy concerns (see H3). There is a correlation by definition. The theoretical framework is inadequate. Think clearly about what is the dependent variable and what is the independent variable. From my point of view there are too many hypotheses.

Experimental design

PLS-SEM cannot estimate causal relationships. What you estimate are correlations. You need to be more careful in the interpretation of results (e.g., not writing ”x impacts y ”, or ”has effect on ”) and generally create awareness that a correlation does not necessarily indicate causation. Apart from acknowledging that endogeneity might be present, there are also ways to deal with endogeneity, such as instrumental variables, control function approaches, propensity score matching, experiments and natural experiments. Hair et al. (2020) note that four conditions must be met to assume causality: (1) Time sequence —the cause must occur prior to the effect; (2) Covariance —changes in the hypothesized independent variable are associated with changes in the dependent variable; (3) Non-spurious associations — the relationship is not due to other variables that may affect cause and effect; and (4) Theoretical support — a logical explanation for the relationship.

Reference
Hair, J. F., Jr , Howard, M. C. , & Nitzl, C. (2020a). Assessing measurement model quality in PLS-SEM using confirmatory composite analysis. Journal of Business Research, 109 , 101–110 .
Pesämaa, O., Zwikael, O., HairJr, J., & Huemann, M. (2021). Publishing quantitative papers with rigor and transparency. International Journal of Project Management, 39(3), 217-222

Validity of the findings

Method: PLS-SEM is used. Yes it is correct that PLS-SEM is estimated by least squares. Please add information on the stata command. Robustness checks are missing. Please report some validity tests (Degree of correlation between constructs, Path significance -Assesses how strongly two constructs relate to each other, Coefficient of determination - Measure of the predictive accuracy of the structural model, Effect size- Measure of the impact of the exogenous construct on the endogenous one, Predictive relevance - Measure of a model's predictive power and Robustness checks).
The article needs to be thoroughly edited and proofread by a qualified language editor. There is an illogical flow of ideas and a lack of coherence. Sections are disjointed without any link between them.

Additional comments

no comment.

---

## Round 0.2 · Minor Revisions

Reviewers have commented about the hypothesis and also about the validity of the recommendations statistically. In particular include the statistical measures that are considered during the experimental analysis.

Reviewer 1 ·

Basic reporting

no comment

Experimental design

no comment

Validity of the findings

no comment

Additional comments

no comment

·

Basic reporting

The comments are given in the previous review are addressed by the authors

Experimental design

Ok

Validity of the findings

Ok

Additional comments

Nil

·

Basic reporting

Some of the hypotheses are trivial and self-explanatory: check them out: Perceived security awareness has a higher awareness of the security and privacy settings of video conferencing apps. I suggest to de-emphasise this.
.

Experimental design

"We used a survey experiment to determine "
Please rewrite. This paper applied PLS-SEM to a stand alone survey. This is not an experimental design.

Validity of the findings

PLS-SEM
Please report all the needed validity tests for the measurement and structural model. This part is not complete.

Additional comments

Please follow the style guidelines
Alammary, A., A. Carbone, and J. Sheard. 2016. “Blended Learning in Higher Education: Delivery Methods Selection.” In Twenty-Fourth European Conference on Information Systems (ECIS 2016). İstanbul,Turkey.
Alammary, Ali, Moneer Alshaikh, and Areej Alhogail. 2021. “The Impact of the COVID-19 Pandemic on the Adoption of e-Learning among Academics in Saudi Arabia.” Behaviour & Information Technology, September, 1–23. https://doi.org/10.1080/0144929x.2021.1973106

---

## Round 0.3 · Minor Revisions

I am glad to inform you that your manuscript is nearly ready for acceptance. However, as indicated by the reviewer I suggest you to add the future scope and limitations in the manuscript. Also, thoroughly check the language and try to improve the presentation of the article.

·

Basic reporting

The revised version looks good to me
Conclusions
Please provide information on the limitations
The text is not always smooth. The language can be improved.
See an example:
Privacy concerns are significantly associated with higher awareness only for STEM faculty, while attitudes toward ICT for teaching and learning is negatively associated with such awareness among faculty with more than 10 years of experience.

->
Privacy concerns are significantly associated with higher awareness only among teachers of STEM subjects, while attitudes towards ICT for teaching and learning are negatively associated with such awareness among teachers with more than 10 years of experience.

Experimental design

PLS-SEM is standard. All tests are reported

Validity of the findings

OK

---

## Round 0.4 · accepted · Accept

I am pleased to inform you of the acceptance decision on your manuscript.